# Development and Characterization of Recombinase-Based Isothermal Amplification Assays (RPA/RAA) for the Rapid Detection of Monkeypox Virus

**DOI:** 10.3390/v14102112

**Published:** 2022-09-23

**Authors:** Lingjing Mao, Jiaxu Ying, Benjamin Selekon, Ella Gonofio, Xiaoxia Wang, Emmanuel Nakoune, Gary Wong, Nicolas Berthet

**Affiliations:** 1Centre for Microbes, Development, and Health, Institut Pasteur of Shanghai, Chinese Academy of Sciences, Unit of Discovery and Molecular Characterization of Pathogens, Shanghai 200031, China; 2University of Chinese Academy of Sciences, Beijing 100049, China; 3Institut Pasteur of Bangui, Bangui, Central African Republic; 4School of Public Health, Lanzhou University, Lanzhou 730000, China; 5Viral Hemorrhagic Fevers Research Unit, CAS Key Laboratory of Molecular Virology and Immunology, Institut Pasteur of Shanghai, Chinese Academy of Sciences, Shanghai 200031, China; 6Institut Pasteur, Unité Environnement et Risque Infectieux, Cellule d’Intervention Biologique d’Urgence, 75724 Paris, France

**Keywords:** monkeypox virus, nucleic acid detection, recombinase polymerase amplification assay, CRISPR-Cas12a, rapid detection system, lateral flow

## Abstract

Monkeypox is a zoonotic disease caused by monkeypox virus (MPXV), in which outbreaks mainly occurred in West and Central Africa, with only sporadic spillovers to countries outside Africa due to international travel or close contact with wildlife. During May 2022, multiple countries in Europe, North and South America, Australia, Asia, and Africa reported near-simultaneous outbreaks of MPXV, the first time that patient clusters were reported over such a large geographical area. Cases have no known epidemiological links to MPXV-endemic countries in West or Central Africa. Real-time PCR is currently the gold standard for MPXV diagnostics, but it requires trained laboratory personnel and specialized equipment, and results can only be obtained after several hours. A rapid and simple-to-operate point-of-care diagnostic test for MPXV is crucial for limiting its spread and controlling outbreaks. Here, three recombinase-based isothermal amplification assays (RPA/RAA) for the rapid detection of MPXV isolates were developed. These three assays target the MPXV G2R gene, and the limit of detection for these systems is approximately 10^0^ copies of DNA per reaction. The assays were found to be specific and non-cross reactive against other pox viruses, such as vaccinia virus, and the results can be visualized within 20–30 min. The assays were validated with DNA extracted from 19 clinical samples from suspected or confirmed MPXV patients from Central Africa, and found to be consistent with findings from traditional qPCR. These results provide a solid platform for the early diagnosis of potential MPXV cases, and will help with the control and prevention of current and future outbreaks.

## 1. Introduction

Monkeypox is a viral zoonotic disease caused by monkeypox virus (MPXV) [1], and has a similar clinical presentation to smallpox, but manifests as a milder form with lower fatality rates in humans (between 1 to 10%). MPXV belongs to the genus, Orthopoxvirus, which is the largest known enveloped double-stranded DNA virus [1]. MPXV is subdivided into two clades: Congo Basin (clade 1) and West African (clade 2). The Congo isolates are known to be more infectious and cause more severe disease, with a mortality rate of ~10% [2]. MPXV was first discovered and isolated in 1958 from colonies of cynomolgus monkeys (*Macaca fascicularis*) imported into Copenhagen, Denmark; thus, the name, “Monkeypox virus” [3]. Human infection with MPXV was first reported in 1970, in which the virus was successfully isolated from a nine-month-old child who had a smallpox-like disease in the Democratic Republic of the Congo (DRC) [4]. Cases have since been subsequently reported in West Africa and especially in Central Africa. Indeed, over the past two decades, the DRC and the Central African Republic (CAR) are the two main countries in the Congo Basin region that have reported the most cases of MPXV. Genomic analyses of viral sequences from these cases have allowed for hypotheses to be made about the relationships of the cases to each other and their origins. For example, phylogenetic analyses of MPXV strains isolated between 2001 and 2018 in CAR indicated that all isolates originated in the Congo Basin and evolved from at least three ancestral isolates originating in the DRC [5]. This suggests that there have been multiple exports of MPXV from the forests of the Congo Basin to CAR, and that the virus likely came from wild animals living in these forests that sporadically transmitted it to humans [5].

With the eradication of smallpox in 1980 and the subsequent cessation of smallpox vaccinations worldwide, MPXV has emerged as the primary orthopoxvirus of public health concern [6]. The first MPXV cases outside Africa were reported during 2003 in the USA, in which the outbreak was associated to contact with sick prairie dogs that were kept or sold as pets [7]. Since the 1990s, the frequency, scale, and secondary transmission events of monkeypox outbreaks have increased. For instance, a total of 502 cases and 8 deaths were documented in Nigeria between 2017 to 2019 [8,9]. Subsequently, several human cases related to the Nigeria outbreak have been reported outside the African continent. Three human cases were reported in the UK during 2018, two of which had travel history in Nigeria [10]. The third case was a hospital worker who was infected via nosocomial transmission, constituting the first confirmed case of human-to-human MPXV transmission outside of Africa. Following these reports from the UK, imported cases were also reported in Israel, Singapore, and other countries [11,12]. The Israeli case had returned from a trip to Nigeria in 2018, whereas a Nigerian patient traveled to Singapore in 2019. During May 2022, multiple countries in Europe, North America, South America, Africa, Australia, and Asia reported near-simultaneous outbreaks or importations of monkeypox. As of 5 August 2022, over 28,000 cases have been confirmed, impacting over 80 countries. It is the first time that monkeypox cases and clusters have been reported over such a large geographical area, in which patients had no known epidemiological links to MPXV-endemic countries in West or Central Africa. The 2022 MPXV isolates were found to belong phylogenetically to the West African clade. Analysis of the virus genomes showed that there was an extremely close relationship between the viral sequences from patient samples, supporting the notion of sustained human-to-human transmission of MPXV during this epidemic [13].

MPXV is known to infect a wide range of mammalian species, but the natural reservoir host and maintenance cycle of the virus in nature remains poorly characterized; although, terrestrial and arboreal rodents are considered to be the source of transmission to humans [14]. Indeed, live MPXV has only been isolated rarely in wild fauna: once from the rope squirrel (*Funisciurus anerythrus*) in Yambuku, DRC in 1985 [15], and once from the sooty mangabey (*Cercocebus atys*) in the Ivory Coast in 1992 [16]. At the Thai National Park in the Ivory Coast in 2020, MPXV was detected in chimpanzee feces during an epizootic event [17]. Finally, MPXV was also detected in different animal hosts (two species of squirrels and rats, and a shrew) in the DRC [18]. The monkeypox outbreak in the USA during 2003 was initiated by the importation of African rodents belonging to three genera: *Graphiurus*, *Cricetomys*, and *Funisciurus* (African dormice, giant pouched rat, and rope squirrel, respectively) [1,15,19]. Zoonotic spillover events are more likely in remote locations of Central and West Africa, where changing ecological factors, exacerbated by explosive demographic changes, rapid urbanization, and the destruction of natural habitats, are bringing human populations closer and in more frequent contact with suspected animal reservoirs [20].

The detection of MPXV nucleic acid is critical for surveillance and diagnostic efforts, which help reduce virus transmission by positively identifying infected cases. Several diagnostic methods, including virus isolation, electron microscopy, and polymerase chain reaction (PCR), have been used to diagnose monkeypox. The gold standard for MPXV detection is real-time PCR, due to its high sensitivity and specificity [21]. However, an equipped laboratory and trained technicians are needed to perform real-time PCR, which are not always available in resource-poor areas where MPXV is endemic, thus limiting its utility in point-of-care applications. Therefore, a simple point-of-care test is crucial for the rapid and convenient diagnosis of MPXV infections.

Clustered regularly interspaced short palindromic repeats (CRISPR) systems are a fundamental part of the microbial adaptive immune system that recognizes foreign nucleic acids based on their sequence, and then subsequently eliminates them by means of endonuclease activity associated with the CRISPR-associated proteins (Cas) enzyme [22]. CRISPR-Cas-targeted recognition and nonspecific cleavage activities have shown promise for the development of in vitro nucleic acid detection technologies [23,24,25,26]. CRISPR-Cas systems, such as CRISPR-Cas9 and CRISPR-Cas12, are widely used in gene editing and gene therapy [27]. CRISPR-Cas12a proteins are RNA-guided DNA targeting enzymes that bind and cut DNA as components of bacterial adaptive immune systems [28]. In the CRISPR-Cas12a detection system, the complementary crRNA guides Cas12a to target dsDNA, activating the subsequent cleavage of short ssDNA reporters carrying a fluorophore [23].

To improve the sensitivity, DNA amplification is essential to most nucleic acid testing strategies. Isothermal amplification methods have been shown to be an alternative to real-time PCR. Recombinase-based isothermal amplification assays, such as recombinase polymerase amplification (RPA), developed by TwistDx (Cambridge, United Kingdom), or recombinase-aid amplification (RAA) by ZC Bioscience (Hangzhou, China), are promising candidates due to their simplicity, high sensitivity, selectivity, compatibility with multiplexing, rapid amplification, as well as their operation at a constant temperature, without the need for an initial denaturation step or the use of multiple primers [29]. The amplification process starts when a recombinase protein binds to primers, forming a recombinase–primer complex. The complex then integrates double-stranded DNA seeking a homologous sequence, and enables the sequence-specific recognition of template target sites by oligonucleotide primers, followed by strand-displacing DNA synthesis, thus resulting in the exponential amplification of the target region within the template. The whole reaction can be performed at constant temperature (optimally around 37–42 °C). For real-time detection, a fluorophore/quencher-probe is used. Furthermore, the RAA or RPA amplification products can then be conveniently and easily detected with lateral-flow strip test technology, often used as a simple, disposable diagnostic device [29]. Thus, the combination of RPA/RAA and CRISPR systems show promise in point-of-care pathogen diagnostics without the need for a well-equipped laboratory, and have already been adapted for the detection of pathogens such as cutaneous and visceral leishmaniasis, intestinal protozoa, or SARS-CoV-2 [30,31,32].

In this study, we developed and validated three recombinase-based isothermal amplification assays: RPA combined with CRISPR-Cas12a (RPA-Cas12a), real-time RPA, and RAA combined with lateral flow strips (RAA-LFS) against MPXV. These assays work by targeting the tumor necrosis factor (TNF) binding protein gene, which is present in duplicate as ORFs, G2L and G2R, in the inverted terminal repeats of the MPXV genome. These methods were found to produce reliable diagnostic results within 20–30 min, visualized as either fluorescence in tubes or as a band on the strip test, and the results are presented as follows.

## 2. Materials and Methods

### 2.1. Isolation of Monkeypox Virus and DNA Extraction

A total of 19 biological samples were collected from patients who presented with skin pustules during previous MPXV outbreaks in CAR. For 3 patients, MPXV was successfully isolated and amplified after intracerebral inoculation in newborn mice using harvested crusts, as described previously [33]. The brains of these inoculated, dead mice were harvested and resuspended in PBS prior to DNA extraction. DNA extraction was performed both from this suspension and directly from crusts, pus, or serum samples using the QIAamp DNA Mini Kit (Qiagen, Germany), following manufacturer instructions. After this step, the extracted DNA were quantified using the Qubit dsDNA High-Sensitivity Assay with the Qubit 2.0 fluorometer (Life Technologies, Carlsbad, CA, USA), precipitated, and then stored at −20 °C until use.

### 2.2. Preparation of Vaccinia Virus DNA and Varicella-Zoster Virus DNA

Vaccinia virus (VACV) (modified vaccinia virus Ankara strain) was obtained from ATCC (VR-1566), and propagated in Syrian hamster kidney cells (BHK-21 cells). The cell suspension was harvested when the cytopathic effect was observed in 70~80% of infected cells, and lysed by performing freeze–thaw cycles three times. The supernatants containing the virus were then collected by centrifugation at 4000× *g* for 5 min at 4 °C to eliminate infected cell debris. Viral DNA were extracted using a MiniBEST Viral RNA/DNA extraction kit (TaKaRa, Beijing, China), and quantified using a Nanodrop 2000 spectrophotometer. Varicella-zoster virus (VZV) was provided by IPB in CAR, and collected from the crust of the lesions. The phenol–chloroform method was employed to isolate nucleic acid.

### 2.3. Real-Time PCR

The MPXV real-time PCR detection was designed to target the G2R gene using primer and probe sequences, as previously described [34], by using the NovoStart Probe qPCR SuperMix kit and QuantStudio1 (ABI, New York, NY, USA). Reactions were conducted in a 20 μL volume following kit instructions. Reaction cycle parameters were set as follows: 50 °C for 2 min, denaturation at 95 °C for 10 min, followed by 40 cycles of amplification at 95 °C for 15 s and at 60 °C for 1 min.

### 2.4. Molecular DNA Standards, RPA Oligonucleotide, and crRNA Preparation

To verify the assays, the sequence targeting the G2R gene was synthesized and inserted into the pUC57 plasmid vector (Sangon, Shanghai, China). For each assay, the forward primers (F), reverse primers (R), and probes were designed (Table 1). The MPXV-RPA-F primer was used for the three assays, whereas the MPXV-RPA-R primer was only used for real-time RPA and RPA-Cas12a detection. The oligonucleotide sequence used as the template for crRNA transcription (5′-TAATTTCTACTAAGTGTAGATcaggcttgtctaagttgtaacgg-3′) was synthesized by GenScript. The oligonucleotides contained a T7 promoter (capital letters), which was used as the template for in vitro transcription at 37 °C for 16 h using a HiScribe T7 Quick High Yield RNA Synthesis Kit (NEB, Ipswich, MA, USA). The small letters in the crRNA sequence indicate the MPXV target gene. The RNA was purified by VAHTS RNA Clean Beads (Vazyme, Nanjing, China), and quantified by Nanodrop. The ssDNA reporter (5′-FAM-TTTTT-3′BHQ) was used for CRISPR-Cas12a detection. The primers and probes were diluted in nuclease-free water to working concentrations before use.

### 2.5. Real-Time RPA

Real-time RPA was carried out with the TwistAmp Exo Kit (TwistDx, Cambridge, United Kingdom). The stock reaction mix was prepared as follows (working concentrations in parentheses): 29.5 μL rehydration buffer, 1 μL of extracted DNA template, 2.1 μL forward primer (10 μM), 2.1 μL reverse primer (10 μM), 0.6 μL probe (10 μM), 12.2 μL dH_2_O, and 2.5 μL magnesium acetate (280 mM). All reagents, except for the template or sample DNA and magnesium acetate, were prepared as a master mix, and added into a 0.2 mL tube containing a dried enzyme pellet. After adding 1 μL of plasmid template or sample DNA to the master mix tube, magnesium acetate was then pipetted into the tube lids, spun down briefly, and mixed well to start the reaction. The reaction mixture was first treated in a MiniAmp thermocycler at 37 °C for 4 min (brief mix, centrifugation, and vortex) and then incubated for 20 min at 37 °C, during which, the fluorescence (FAM) signal was collected every 30 s with QuantStudio1 (ABI, United States).

### 2.6. RPA Combined with CRISPR-Cas Detection

The RPA-Cas12a detection platform included two individual steps: the RPA amplification and the CRISPR-Cas12a-mediated cleavage assay. The RPA basic kit (TwistDx, Cambridge, United Kingdom) was first used to amplify the nucleic acid. The RPA step was performed in a 50 μL volume. The stock reaction mix included 29.5 μL rehydration buffer, 1 μL template or extracted DNA sample, 2.4 μL forward primer (10 μM), 2.4 μL reverse primer (10 μM), 12.2 μL dH_2_O, and 2.5 μL magnesium acetate (280 mM). The reaction mixture was first treated in a MiniAmp thermocycler at 37 °C for 4 min (brief mix, centrifugation, and vortex), and then incubated for 20 min at 37 °C to generate the amplified product. For CRISPR-Cas12a detection, the reaction mix consisted of 1.3 μL Cas12a (1.67 μM), 1 μL crRNA (2.7 μM), 0.5 μL ssDNA reporter (2 μM), 2 μL NEB Buffer (10×), 14.2 μL nuclease-free water, and 1 μL of the amplified product. The final ratio of Cas12a: crRNA: ssDNA reporter was 4:5:2. The final reaction was incubated at 37 °C for 25 min, and the fluorescence (FAM) signal was detected every 30 s with QuantStudio1 (ABI, United States).

### 2.7. RAA Combined with Lateral Flow Assays

RAA-LFS consisted of two steps, the RAA amplification and the visualization of the results by lateral flow test strips (Milenia Biotec GmbH, Giessen, Germany). The RAA assay was performed in a 50 μL volume using the RAA-nfo nucleic acid amplification kit (ZC Bioscience, Hangzhou, China). Each reaction included 40.9 μL rehydration buffer, 2 μL template or extracted DNA sample, 0.4 μL forward primer (10 μM), 0.4 μL reverse primer (10 μM), 0.12 μL probe (10 μM), 3.68 μL dH_2_O, and 2.5 μL magnesium acetate (280 mM). All reagents, except for the template or sample DNA and magnesium acetate, were prepared as a master mix, and added into a 0.2 mL tube containing a dried enzyme pellet. After adding 2 μL of plasmid template or sample DNA to the master mix tube, magnesium acetate was then pipetted into the tube lids, spun down briefly, and mixed well to start the reaction. The reaction mixture was incubated at 37 °C for 10 min. After incubation, 2 μL of the reaction was removed and mixed with 98 μL of HybriDetect assay buffer. Then, 8 μL of the diluted sample was pipetted directly onto the sample application area, and the strips were placed with the sample application area facing downwards into 100 μL HybriDetect Assay Buffer, and incubated for 5–15 min at room temperature in an upright position. If both the test and control bands are displayed, it is a valid positive result. If only the control band is displayed, it is considered to be a valid negative result. Moreover, if the control band is not visible after the incubation period, the result is invalid, and the test must be repeated with a new strip.

## 3. Results

### 3.1. Assay Sensitivity and Specificity

We characterized the sensitivity of all three assays with serial tenfold dilutions of the MPXV DNA plasmid standards (ranging from 10^5^ to 10^0^ DNA copies/μL), and each test was performed in triplicate. The standard curve for quantifying the DNA plasmid amounts was generated by real-time PCR results, and are shown in Appendix A. The fluorescence of the real-time RPA and RPA-Cas12a systems was detected every 30 s for a total of 50 cycles, or 25 min in other words (Figure 1A,B). For the real-time RPA assay, DNA copy numbers as low as 1 copy can result in a significant increase in fluorescence values after 10 min, or 20 cycles. For the RPA-Cas12a assay, a significant increase in fluorescence values can be observed within 5 min, or 10 cycles. The limit of detection for the real-time RPA and RPA-Cas12a is 10^0^ DNA copies per reaction for both assays (Figure 1C,D). The sensitivity of RAA-LFS was also tested with the same MPXV DNA plasmid standard series as described above, in which detection was successful for DNA copy numbers as low as 10^0^ DNA copy/μL per reaction (Figure 2A), with clearly visible test and control lines. For all three assays, the non-template control (nuclease-free H_2_O) showed negative detection results (Figure 1 and Figure 2A). The specificity of the three assays was tested using 28 ng (about 10^9^ copies) DNA of vaccinia virus, and 0.238 ng (about 3.8 × 10^5^ copies) DNA of Varicella-zoster virus, and found to generate negative detection results (Figure 2A, Table 2).

### 3.2. Assay Performance on Clinical Samples as Compared to Traditional Real-Time PCR

To validate the utility of all three MPXV rapid detection assays, 19 DNA samples (18 positives and 1 negative, as confirmed by traditional real-time PCR) extracted from collected patient samples were tested, and the results were compared with traditional real-time PCR (Table 2). The positive and negative thresholds for the real-time RPA and RPA-Cas12a assays were based on the negative control (NC) for that specific experiment, and were defined as follows: if the fluorescence difference between cycle 1 and 50 for the sample was over five times greater than that of the NC, the sample is defined to be positive. If the fluorescence difference for the sample is between two and five times greater than that of the NC, the sample is defined to be borderline positive, and the assay is repeated again. If the fluorescence difference for the sample is less than two times greater than that of the NC, the sample is defined to be negative. The positive and negative thresholds for the RAA-LFS assay were based on visualization with the naked eye. Real-time PCR, real-time RPA, and RPA-Cas12a were performed four times each on all samples, except for the following: three samples (Sample 2, 18, 19) with a Ct value greater than 37 were tested six times with each assay. The RAA-LFS was tested in triplicate with all samples. Compared to real-time PCR, the consistency of results between real-time RPA and RAA-LFS assay for the identification of MPXV was 100% and 100%, even for samples with high Ct values (greater than 37), whereas those for RPA-Cas12a were 83.33% (15/18) for all samples, including those with high Ct values. Therefore, the results suggest that real-time RPA and RAA-LFS may be more sensitive compared to the RPA-Cas12a detection system.

## 4. Discussion

MPXV is a neglected tropical pathogen that has been demonstrated to transmit between humans and rodents, as well as between humans. Transmission occurs by direct contact with infected bodily fluids or lesions, via infectious fomites, or through respiratory secretions, typically requiring prolonged contact [35]. Close contact with infected persons or fomites (e.g., shared linens) is thought to be the most significant risk factor for MPXV infection [36,37,38]. During previous outbreaks, case numbers were typically low and the geographical size of affected areas was small, such as in remote, isolated villages. As such, the rapid spread of MPXV during the epidemic in 2022, with over 28,000 confirmed cases, has taken authorities by surprise, and the WHO subsequently declared MPXV as a public health emergency of international concern (PHEIC) in July 2022 [39]. It appears that a substantial number of cases in the 2022 epidemic involve men who have sex with men (MSM) [40,41]. It is not yet known whether monkeypox can be transmitted through semen or other bodily fluids, but transmission has been known to occur from close skin-to-skin contact. However, additional investigations are needed to comprehensively understand the exact modes of transmission for MPXV. To control the number of cases and prevent further spread, it is crucial to accurately diagnose MPXV and distinguish confirmed cases from infection with other pathogens that may cause similar clinical symptoms, such as chickenpox, measles, molluscum contagiosum, or some rickettsioses [42,43]. During the national surveillance of monkeypox in the DRC, most suspected cases were in fact chickenpox [44], which is a common, highly contagious illness caused by Varicella-zoster virus (VZV), belonging to the family, Herpesviridae. Chickenpox is usually a self-limiting disease lasting 4–5 days with fever, malaise, and a generalized vesicular rash of blister-like lesions. The rash covers the entire body, but is usually more concentrated on the face, scalp, and trunk [45]. Primary infection (chickenpox) is most often seen in children, and rarely in adults. Monkeypox have similar clinical manifestations as chickenpox, but the disease can prove fatal. Mortality is higher among children and adults with comorbidities, and the disease course is more severe in immunocompromised individuals [14]. The epidemiologic characterization and clinical identification of monkeypox in the DRC is also complicated by the coendemicity of both viruses, which make differential diagnosis between monkeypox and chickenpox particularly challenging in the field [42,43,46]. Therefore, the development of a rapid test that can be used for clinical and field evaluation is key for helping community health workers to distinguish between these two conditions, and to perform surveillance studies in humans, as well as any potential reservoir animal species.

In this study, we developed three isothermal, rapid, and reliable recombinase-based rapid detection methods for MPXV, which generates accurate results within 20–30 min. The real-time RPA and RAA-LFS results showed 100% consistency with real-time PCR results for MPXV detection, whereas RPA-Cas12a showed 83.3% consistency. RPA-Cas12a was performed in two separate steps: the isothermal amplification step is performed first, after which, a small amount of amplification product is transferred to the CRISPR-Cas detection system. During the amplification of low concentrations of nucleic acid, localized rapid amplification may occur when transferring a small amount of amplification product to a new tube, causing sampling errors, and resulting in reduced detection sensitivity. A one-tube detection platform based on RT-RPA and CRISPR/Cas12a has been reported to detect SARS-CoV-2, and showed 100% consistency with rRT-PCR [47], and can reduce the risk of producing potential contamination in the lab. For future studies, it is necessary to further optimize the RPA-Cas12a method to a one-tube detection in order to improve the sensitivity. For the real-time RPA, the generation of fluorescence signals depends on the probe binding to the RPA amplification product. As a DNA repair enzyme, exo recognizes the internal empty base site (THF) and cleaves the probe to separate the fluorophore from the quencher to generate fluorescence. The products can also be detected via lateral flow with a strip test [29]. This assay includes both a fluorophore-labeled internal probe, as well as a reverse primer that is biotinylated at the 5′ end. When the amplification product containing the labeled probe interacts with gold particles labeled with the mouse anti-fluorophore antibody, only amplicons containing the biotinylated reverse primer are immobilized by the anti-biotin antibodies on the corresponding band. However, the test is only validated when the control band, which is located in the upper part of the strip, also appears, due to the immobilization of excess free gold particles [48].

Isothermal amplification assays based on loop-mediated isothermal amplification (LAMP) technology have previously been developed for MPXV [49]. Though this assay also negated the need for specific laboratory equipment, the assay was found to have 72% consistency compared with the results from nested PCR, and approximately 60 min was needed to generate the results. Additionally, the LAMP MPXV assay requires six pairs of primers, but the three recombinase-based amplification assays only need one pair of primers and one probe, thus simplifying the diagnostic design [49]. Previously published findings on an MPXV-RPA assay targeting the G2R gene are consistent with our real-time RPA results, in which the limit of detection was determined to be 16 DNA molecules/μL, and which produced diagnostic results within 3 to 10 min [50]. The results were verified by fluorescence values, which required specialized instruments and trained laboratory staff, limiting their utility for home use by non-professionals. In contrast, RAA-LFS is much more user-friendly due to its simplistic operation and similarity to other strip-based assays, such as pH or pregnancy tests, which make self-testing at home more feasible.

## 5. Conclusions

In conclusion, these results summarize the adaptation and combination of RPA/RAA and CRISPR-Cas technologies for the rapid, accurate, and convenient detection of MPXV, highlighting their promising role in the prevention and control of outbreaks in endemic and non-endemic regions.

## Figures and Tables

**Figure 1 viruses-14-02112-f001:**
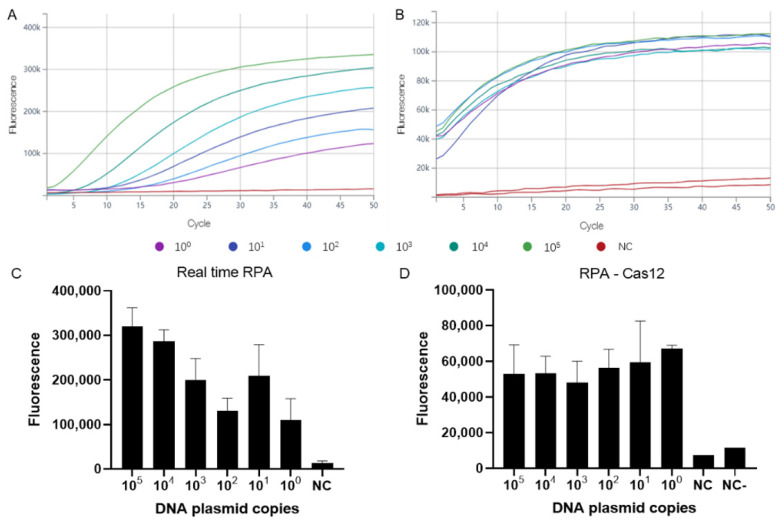
Sensitivity of the real-time RPA and RPA-Cas12a detection systems. Serial 10-fold dilutions of MPXV DNA plasmids (10^5^–10^0^ copies) were detected by real-time RPA (**A**), or amplified by RPA and detected by the CRISPR-Cas12a detection system (**B**). Each cycle represents 30 s. Fluorescence values of real-time RPA (**C**) and RPA-Cas12a after 50 cycles (**D**). NC (negative control) denotes reaction without DNA, NC—denotes reaction without crRNA.

**Figure 2 viruses-14-02112-f002:**
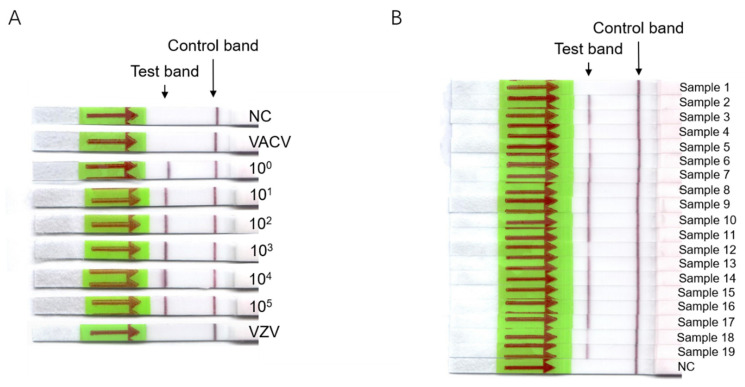
Sensitivity and specificity of RAA-LFS assays. Serial 10-fold dilutions of MPXV DNA plasmids (10^5^–10^0^ copies) were detected by RAA-LFS assays (**A**). Results from 19 DNA samples extracted from clinical specimens from Central Africa are shown (**B**). NC denotes negative control. VACV denotes vaccinia virus. VZV denotes Varicella-zoster virus. Red arrows and green show the flow through direction.

**Table 1 viruses-14-02112-t001:** Primers and probes for different assays. FHQ represents the sites of the quencher and fluorophore in the following order: Fam-dT (F), Tetrahydrofuran (H), and BHQ1-dt (Q). BLOCK is the polymerase extension blocking group.

Name	Sequence (5′ to 3′)	Assays
MPXV-RPA-F	ATCCAATGGAAAATGTAAAGACAACGAATACAG	Real-time RPA, RPA-Cas12a, RAA-LFS
MPXV-RPA-R	TCGTGTTACACGATCGCGTCTCTACCTGATTA	Real-time RPA, RPA-Cas12a
MPXV-exo-Probe	GTGATAGCAAGACTAATACACAATGTACGCCG**FHQ**GGTTCGGATACCTTT**-BLOCK**	Real-time RPA
MPXV-nfo-Probe	**6-FAM**-GTGATAGCAAGACTAATACACAATGTACGCCGT**H**TGGTTCGGATACCTTT**-BLOCK**	RAA-LFS
MPXV-nfo-R	**Biotin**-TCGTGTTACACGATCGCGTCTCTACCTGATTA	RAA-LFS
MPXV-PCR-F	GGAAAATGTAAAGACAACGAATACAG	Real-time PCR
MPXV-PCR-R	GCTATCACATAATCTGGAAGCGTA	Real-time PCR
MPXV-PCR-Probe	AAGCCGTAATCTATGTTGTCTATCGTGTCC	Real-time PCR

**Table 2 viruses-14-02112-t002:** Detection of MPXV using the three recombinase-based isothermal amplification assays in clinical samples, and compared to real-time PCR. NC: negative control; VACV: vaccinia virus; VZV: Varicella-zoster virus; +: positive; −: negative; unless otherwise stated, the samples were tested four times, and + or − indicate consistent results between all replicates. The numbers in parentheses indicate the amount of positive results out of the total number of replicates.

Sample ID	Sample Type	Assay Results			
		Real-Time PCR (Average Ct for Positive Results)	Real-Time RPA	RPA-Cas12a	RAA-LFS
1	Pus	Negative	-	-	-
2	Crusts	37.95451 (4/6)	+ (4/6)	− (2/6)	+ (2/3)
3	Mice brain biopsy (from crusts)	20.68634	+	+	+
4	Pus	33.41898	+	+	+
5	Crusts	26.33024	+	+	+
6	Mice brain biopsy (from crusts)	25.58751	+	+	+
7	Crusts	31.04101	+	+	+
8	Mice brain biopsy (from crusts)	23.25146	+	+	+
9	Pus	22.62705	+	+	+
10	Pus	25.66506	+	+	+
11	Pus	30.93425	+	+	+
12	Pus	22.09036	+	+	+
13	Pus	24.65622	+	+	+
14	Pus	19.49808	+	+	+
15	Pus	23.24219	+	+	+
16	Pus	23.97732	+	+	+
17	Serum	31.4181	+	+	+
18	Serum	38.22707 (4/6)	+ (5/6)	-	+ (2/3)
19	Serum	37.72934 (2/6)	+ (5/6)	-	+ (2/3)
NC		Negative	-	-	-
VACV		Negative	-	-	-
VZV	Crusts	Negative	-	-	-

## Data Availability

Not applicable.

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
