# Peer review of "Development and Characterization of Recombinase-Based Isothermal Amplification Assays (RPA/RAA) for the Rapid Detection of Monkeypox Virus"

_viruses, 2022, doi:10.3390/v14102112_

Round 1

Reviewer 1 Report

In this manuscript titled "Development and Characterization of Recombinase-Based Isothermal Amplification Assays (RPA/RAA) for the Rapid Detection of Monkeypox virus", the authors explored three Recombinase-Based Isothermal Amplification Assays for detection of MPXV, and found that these assays had a higher specificity and a high sensitivity. Overall, this is an interesting and timely study. However, some controls were missing or not well explained. Therefore, a major revision is needed.

Major issues:

1. Please give some more introduction of the rationale of Recombinase-based isothermal amplification and how it works.

2. In the figure legend of Figure 1, "NC denotes negative control", this looks vague. Does the negative control mean no DNA at all or with DNA of vaccinia virus? Actually, both negative controls are needed in this experiment to demonstrate the specificity of this assay.

3. In Table 2, nearly all of the samples were tested as "+" in RPA/RRA assays, but how was the "+" threshold determined in the RPA/RRA assays? 

Author Response

Reviewer 1: In this manuscript titled "Development and Characterization of Recombinase-Based Isothermal Amplification Assays (RPA/RAA) for the Rapid Detection of Monkeypox virus", the authors explored three Recombinase-Based Isothermal Amplification Assays for detection of MPXV, and found that these assays had a higher specificity and a high sensitivity. Overall, this is an interesting and timely study. However, some controls were missing or not well explained. Therefore, a major revision is needed.

Major issues:

  1. Please give some more introduction of the rationale of Recombinase-based isothermal amplification and how it works.
  2. In the figure legend of Figure 1, "NC denotes negative control", this looks vague. Does the negative control mean no DNA at all or with DNA of vaccinia virus? Actually, both negative controls are needed in this experiment to demonstrate the specificity of this assay.
  3. In Table 2, nearly all of the samples were tested as "+" in RPA/RRA assays, but how was the "+" threshold determined in the RPA/RRA assays? 

Response: Thank you for taking the time to review the manuscript and all the helpful comments. The following changes have been made:

  1. The rationale of RPA has been added to the introduction section, see below:

“The amplification process starts when a recombinase protein binds to primers forming a recombinase-primer complex. The complex then integrates double-stranded DNA seeking a homologous sequence, enables the sequence specific recognition of template target sites by oligonucleotide primers, followed by strand-displacing DNA synthesis, and thus resulting in the exponential amplification of the target region within the template. The whole reaction can be done at constant temperature (optimum around 37-42ËšC). For real-time detection, a fluorophore/quencher-probe is used”. See Page 3, lines 132-139.

  1. The figure legend of Figure 1 has been modified to describe the NC in further detail, see below:

“NC (negative control) denotes reaction without DNA, NC- denotes reaction without crRNA.” See Page 7, lines 275-276..

The results of specificity assays with DNA of vaccinia virus were shown in Table 2 and Figure 2A. The specificity control for Figure 1 is not needed, because it is a sensitivity experiment.

  1. How the threshold was determined has been added in the results section, see below:

“The positive and negative thresholds for the real-time RPA and RPA-Cas12 assays were based on the negative control (NC) for that specific experiment and were defined as follows: if the fluorescence difference between cycle 1 and 50 for the sample was over 5 times greater than that of the NC, the sample is defined to be positive. If the fluorescence difference for the sample is between 2-5 times greater than that of the NC, the sample is defined to be borderline positive and the assay is repeated again. If the fluorescence difference for the sample is less than 2 times greater than that of the NC, the sample is defined to be negative. The positive and negative thresholds for the RAA-LFS assay was based on visualization with the naked eye.” See Page 7, lines 282-290.

Reviewer 2 Report

The work is well structured and clearly written. The most relevant recommendation is to cite and discuss other similar works for rapid virus detection (eg. Davi SD, Kissenkötter J, Faye M, et al. Recombinase polymerase amplification assay for rapid detection of Monkeypox virus. Diagn Microbiol Infect Dis. 2019 ; 95 (1): 41-45. Doi: 10.1016 / j.diagmicrobio.2019.03.015).
Cross reaction conditions for further pathogens should also be evaluated or discussed, in the present work only VACV is considered.
The limit of detection is very good.

Author Response

Reviewer 2: The work is well structured and clearly written. The most relevant recommendation is to cite and discuss other similar works for rapid virus detection (eg. Davi SD, Kissenkötter J, Faye M, et al. Recombinase polymerase amplification assay for rapid detection of Monkeypox virus. Diagn Microbiol Infect Dis. 2019 ; 95 (1): 41-45. Doi: 10.1016 / j.diagmicrobio.2019.03.015).
Cross reaction conditions for further pathogens should also be evaluated or discussed, in the present work only VACV is considered.
The limit of detection is very good.

Response: Thank you very much for the positive evaluation and useful comments. According to your suggestion, we have added some experiments and data to enrich the manuscript. Specifically, we used our detection systems to test for assay specificity against Varicella-Zoster Virus (VZV), a herpesvirus infection that causes similar clinical manifestations to MPXV, the results were added to Table 1 and Figure 2A.

The suggested references were already discussed in the initial submitted manuscript.

Round 2

Reviewer 1 Report

All concerns resolved